# Life Cycle Assessment of Nile Tilapia (*Oreochromis niloticus*) Farming in Kenyir Lake, Terengganu

**Hayana Dullah [1,\*]**, **M. A. Malek [2]** and **Marlia M. Hanafiah [3,4]**

1   UNITEN R & D Sdn. Bhd. Universiti Tenaga Nasional, 43000 Kajang, Selangor, Malaysia
2   Institute of Energy Policy and Research (IEPRe), Universiti Tenaga Nasional, 43000 Kajang, Selangor, Malaysia; marlinda@uniten.edu.my
3   Department of Earth Sciences and Environment, Faculty of Science and Technology, Universiti Kebangsaaan Malaysia, 43600 Bangi, Selangor, Malaysia; mhmarlia@ukm.edu.my
4   Centre for Tropical Climate Change System, Institute of Climate Change, Universiti Kebangsaaan Malaysia, 43600 Bangi, Selangor, Malaysia
\*   Correspondence: hayana.dullah@uniten.edu.my; Tel.: +603-8921-2020

**Abstract:** This study presents results from a life cycle assessment (LCA) conducted following the CML-IA method on caged aquaculture of Nile tilapia (*Oreochromis niloticus*) species at Como River, Kenyir Lake, Terengganu, Malaysia. In this study, the greenhouse gas (GHG) estimation, calculated based on the Intergovernmental Panel on Climate Change (IPCC) 2006 Guidelines, showed that 245.27 C eq (1.69 Kg) of nitrate oxide ($N_2O$) was emitted from the lake. The determination of LCA was conducted using several inputs, namely $N_2O$, compositions of fish feed, materials used to build fish cages (infrastructure), main materials used during operation and several databases, namely Agri-footprint, Ecoinvent 3, European Reference Life-Cycle Database (ELCD), and Industry Data 2.0. The results show that feed formulation is the major contributor to potential environmental impact in aquaculture farming, at 55%, followed by infrastructure at 33% and operation at 12%. The feed formulation consisting of 53% broken rice contributed to marine ecotoxicity (MET), while those consisting of 44% fish meal and 33% soybean meal contributed to abiotic depletion (ABD) and global warming (GW), respectively. It is recommended that the percentage of ingredients used in feed formulation in fish farming are further studied to reduce its impacts to the environment.

**Keywords:** aquaculture; fish farming; environmental impact; life cycle assessment; *Oreochromis niloticus*; Kenyir Lake

## 1. Introduction

Kenyir Lake is an artificial lake created in 1985 by the damming of Kenyir River. The lake provides water to the Sultan Mahmud Power Station, which is operated by Tenaga Nasional Berhad. Since the formation of the lake, the ecological property of forested land slowly developed to conform to a lacustrine environment. This changing ecosystem offers different kinds of services. One of the main activities in Kenyir Lake is freshwater fish farming, i.e., Nile tilapia fish *(Oreochromis niloticus)* located at Como River. To date, the Department of Fisheries (DOF) Malaysia has provided 28,099 hectares for the aquaculture industry across the country [1]. This area involves freshwater fish farming, sea fish farming, cockle farming and hatcheries. Como River has a total area of 2800 hectares for Nile tilapia fish farming [1]. The semi-intensive aquaculture system was adapted for fish farming in Kenyir Lake. Floating net cage aquaculture of Nile tilapia has been implemented since fish farming at Kenyir Lake started in 2007.

The world production for aquaculture has expanded from 1.7 million tonnes in 1957 to 110.2 million tonnes in 2016 [2]. Similarly, Nile tilapia fish production for Kenyir Lake has slightly increased in recent years, from 2011 to 2018, with an average of 145.13 tonnes per year. The increase in the number of operators, fish farm cages and advance production technologies reflects the expansion of fish production yearly [3]. Above all, it involves the increase in production inputs such as feed composition, infrastructure, harvesting tools, energy and chemicals that may negatively affect the environment, hence raising sustainability concerns. In addition, increasing input and output production can potentially cause environmental pollution involving mainly airborne and waterborne emissions from fish farms [4]. Recently, environmental studies of the global scale in many published journals found that various environmental problems, such as ozone layer depletion, global warming and acidification have been found to be caused by aquaculture production [5,6]. For this study, the effects of aquaculture production in a specific area of Kenyir lake were discussed, focusing on the effect of climate change. There are several methods recommended for calculating the sustainability and efficiency of food production systems, including life cycle assessment (LCA), nutrient dynamic modeling and socio-economic analysis [7].

Nile tilapia culturing contributes approximately 90% of the total production of tilapia species in Malaysia [8]. Fish culture activity has been said to be a major contributor to environmental damage due to the increasing production of organic waste and toxic compounds such as polychlorinated biphenyls (PCBs) [9,10]. In China, the environmental impact from aquaculture is mainly due to fish feed and lack of aquaculture management [9]. However, the environmental impacts of aquaculture farming, especially Nile tilapia fish in Malaysia, remain unclear. Since a wide range of chemicals are used in aquaculture systems in Malaysia, it is necessary to identify the hotspots of environmental impact from all the processes involved in order to improve the aquaculture system and maintain environmental sustainability.

To date, several researchers devoted to LCA of fisheries and aquaculture have evaluated a limited number of species. LCA is one of the main assessment tools for evaluating environmental impact, thus identifying the contribution of processes involved in aquaculture production [4]. In the meantime, the mitigation strategies in aquaculture production are a lot easier to implement since LCA provides the hotspots of environmental impact from the process. In accordance with ISO 14040, the LCA methodology consists of four phases in the technical framework; (1) goal and scope, (2) life cycle inventory analysis, (3) life cycle impact assessment and (4) life cycle interpretation. Firstly, defining the goal and scope includes the aim and purpose of study, intended audience, functional unit, system boundary and limitations. Secondly, inventory analysis involves data collection and mass balance calculation for each process in the production system, as well as emission data of air, water and soil. Thirdly, the life cycle impact assessment defines the categories of potential environmental impact used in the system and provides information for the interpretation phase. Lastly, life cycle interpretation is the phase of results from the inventory data being analysed and provides hotspots for environmental impact in the production system involved.

In this study, the environmental implications of cage farming aquaculture in Como River and Kenyir Lake were focused on to examine the quantitative environmental impact in production processes. The system boundary in this study was limited to a farm gate assessment which considered the emissions from farming activity in the Kenyir Lake area. The objectives were to (1) assess the environmental performance of Nile tilapia aquaculture farming in Kenyir Lake by adopting an LCA perspective and (2) evaluate the component from the production system which has the most environmental impact.

## 2. Methodologies

Life cycle assessment (LCA) was chosen as the environmental management tool to evaluate the net cage aquaculture production sectors from a life cycle perspective in the reservoirs of Kenyir Lake, Terengganu. Previous studies have reported significant uses of LCA in the aquaculture system [11–13].

The frameworks of International Standards Organisation (ISO) 14040 and 14044 were used in this study and each phase adopted in this LCA study is explained in the following sections.

*2.1. Study Area and System Description*

The case study of fish farms was located at Como River, Kenyir Lake, Hulu Terengganu, which is one of the production areas for Nile tilapia in Malaysia. The maps location of the area shown in Figure 1 with geographic coordinates of 5°02′09.6″ N 102°50′46.0″ E.

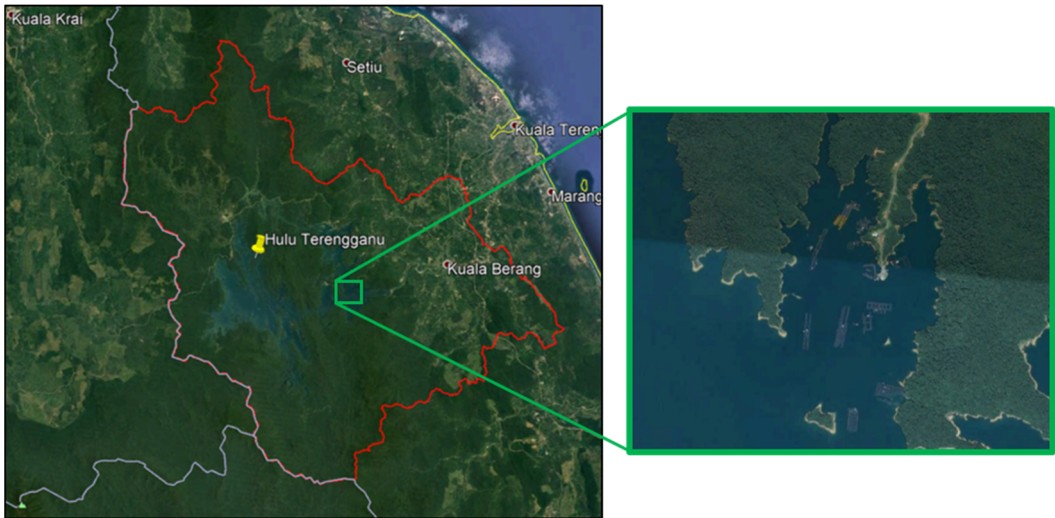

**Figure 1.** Location of Nile tilapia fish farm (right) at Hulu Terengganu.

The primary data in this study were collected from 13 operators of fish farms with 1640 cages. Figure 2 shows the sketch of Nile tilapia cages installed by the operators within the area of fish farming. Each cage with dimensions 6 m × 6 m × 6 m could harvest an average of 600 Nile tilapia fish. The materials and processes of the aquaculture system obtained from the questionnaire and site investigation of the fish farms are elaborated in Section 3.1.

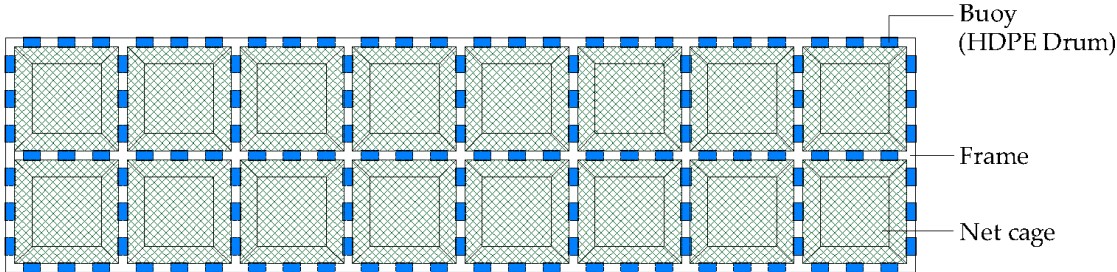

**Figure 2.** Schematic of Nile tilapia cages.

Fingerlings entering the cages with an average weight of 10 g and 3 inches in length are harvested when they reach 500 g after 6 months of rearing. The cages are usually operated all year round with one or two production cycles for Nile tilapia. Initially, 1000 fingerlings are introduced into the cage. Over the duration of the grow-out phase, around 600 fish were taken to be harvested, hence resulting in a mortality rate of about 40%. All farms used commercial pelleted feeds to support fish growth. The feed was distributed manually by cage caretakers on the upper cages, approximately 820 kg per day. The feed conversion rate was 2.1 (FCR = total amount of food given/total amount of fish produced).

## 2.2. Goal, Scope and System Boundary

The first phase in an LCA study is the goal and scope phase. ISO14044 states that the goal of the study "shall unambiguously state the intended application, the reason for the study, and the intended audience" [14]. The main goal of this study was to identify the environmental impacts of rearing Nile tilapia fish on a net cage aquaculture farm. The scope of the study was chosen based on the goal and included the level of detail, systems and processes, functional units, system boundary, impact categories, interpretation methods and allocation methods. The system boundaries in this study included the inputs (fish feeding, infrastructure and operation) and outputs (e.g., GHG emissions, nitrogen (N) and phosphorus (P) emissions). Several stages in Nile tilapia production were excluded from the study due to lack of reliable data and data outside the area of study (e.g., productions, transportation and post-harvest).

A few published LCA studies in aquaculture have been reviewed, as shown in Table 1. The studies compare aquaculture species, functional unit, system boundary, tools and impact. In many countries, environmental impact studies for aquaculture have focused on global warming, eutrophication potential, acidification potential and abiotic depletion.

**Table 1.** General methodology of some life cycle assessment (LCA) studies related to aquaculture.

| Authors, Year | Species | Functional Unit | System Boundary | Software | Impact Category |
|---|---|---|---|---|---|
| **Abdou et al., (2018)** [15] | Seabass and Seabream | 1 tonne of fish | Cradle-to-gate | SimaPro v. 8.0 | ACD, EUT, GW, LOP, NNPU, TCED |
| **Parker (2018)** [16] | Atlantic Salmon | 1 tonne of salmon | Farm gate | SimaPro v. 8.1 | GW, TCED, PO ACD, EUT, OLD, |
| **Yacout et al., (2016)** [17] | Tilapia | 1 tonne of tilapia | Cradle-to-farm-gate | SimaPro v. 7.2 | ACD, EUT, GW, TCED |
| **Pahri et al., (2016)** [18] | Cockles | 70 kg of cockles | Cradle-to-farm-gate | SimaPro v. 8 | ABD, GW, HT, FET, MET, PO, ACD, EUT, OLD, TET |
| **(García et al., 2016)** [19] | Gilthead seabream | 1000 tonnes | Cradle-to-grave | SimaPro v. 8.04 | ABD, GW, OLD, PO, ACD, EUT, TCED |
| **(Ayer et al., 2016)** [20] | Atlantic Salmon | 1 tonne of salmon | Market | SimaPro v. 7.3.3 | ACD, EUT, GW, TCED, MET, MD |
| **(Aubin et al., 2015)** [21] | Black tiger prawn, Mud crabs, Tilapia and Milkfish | 1 tonne of product | Cradle-to-gate | SimaPro v. 7.0 | EUT, ACD, GW, LOP, NNPU, TCED |
| **Dekamin et al., (2015)** [22] | Rainbow trout | 1 tonne of live fish | Farm Gate | SimaPro 7.1 software | EUT, ACD, GW, ABD, HT, FET, PO, WD |
| **Pongpat and Tongpool (2013)** [23] | Nile Tilapia and Striped Catfish | 1 kg of fish | Cradle-to-gate | SimaPro v. 7.3 | ABD, ACD, GW |
| **Mungkung et al., (2013)** [24] | Carp and tilapia | 1 tonne of fish | Cradle-to-farm gate | Not stated | WD, ACD, EUT, GW, LOP, TCED, NPPU |

Notes: abiotic depletion (ABD), acidification (ACD), eutrophication (EUT), global warming (GW), ozone layer depletion (OLD), photochemical oxidation (PO), human toxicity (HT), marine ecotoxicity (MET), terrestrial ecotoxicity (TET), freshwater ecotoxicity (FET), total cumulative energy demand (TCED), net primary production use (NPPU), water dependence (WD), metal depletion (MD), land occupation potential (LOP).

The only LCA on aquaculture study in Malaysia on cockle farming [18] discussed ten environmental impacts.

### 2.2.1. System Boundary

An LCA is performed by defining product systems as models which describe the core of physical systems. The system boundary determines which processes and phases are included in the foreground system and it has to be consistent with the goal and scope of the study [25].

In this study, the LCA was performed to establish the environmental impact of the Nile tilapia production system. The major processes and system boundary considered in this study for the impact evaluation of fish rearing in cages in the area of fish farming are shown in Figure 3. This study focuses

on cage fish farming activities at a lake illustrated as a "system boundary". A farm gate assessment was applied to the case study farms for tilapia production. The system boundaries of this study excluded the production and transportation of fish feed and fingerlings.

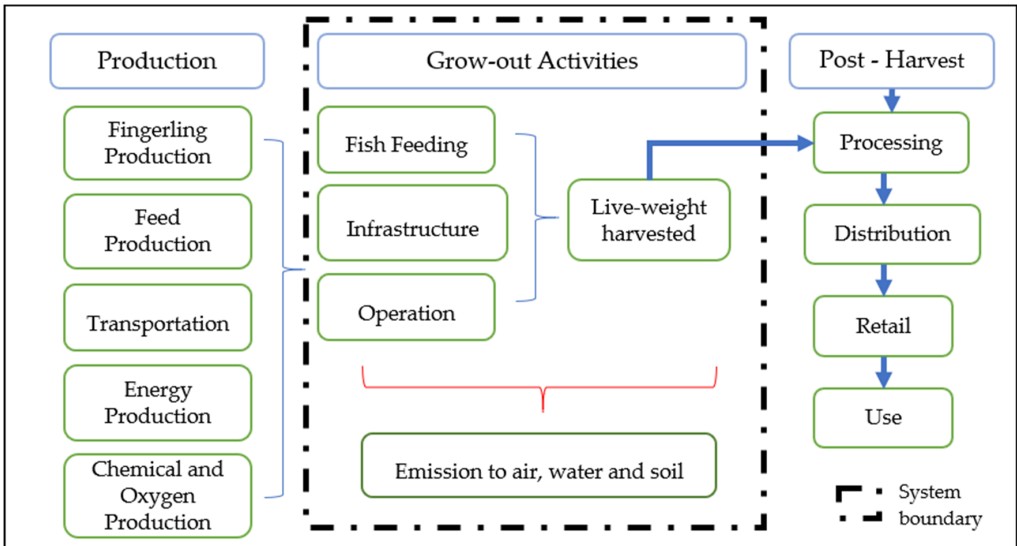

**Figure 3.** Major processes in production of Nile tilapia cage farming.

2.2.2. Functional Unit

A functional unit is the reference unit which the whole study is based upon. It is the reference flow, meaning that all other flows in the modelled system are related to it. It has to be clearly defined and measurable [26]. The functional unit (FU) of the Nile tilapia production system was 1 tonne of live fish harvested [22,27].

## 3. Results

The life cycle inventory and impacts of Nile tilapia cage farming activities at Kenyir Lake to the environment are illustrated in this subsection.

*3.1. Life Cycle Inventory (LCI)*

In this study, the life cycle inventory (LCI) was conducted based on ISO 14041 (1998) [28] using input and output data as listed in Tables 2 and 3. The preparation of inventory data included primary and secondary data. The primary data consisted of questionnaires and laboratory results obtained from water quality samplings collected at the study area. Meanwhile, the secondary data were obtained from companies, manufacturers, authorities, established databases and the literature review.

The input data used in this study are summarised in Table 2. Column 1 is the classification of major activities which contribute to the environmental impact study based on literature reviews. The elements in Column 2 were obtained from onsite questionnaire and surveys from different fish farmers of 13 farms operating on the Como River and literature reviews. The data of compositions in Column 3 were obtained from companies, manufacturers, authorities, established databases and literature reviews. The percentages of feed ingredients were obtained from the fish feed supplier, while the quantity was calculated based on the usage per tonne of harvested fish. The amount of gases emitted from the processes involved during the manufacturing of compositions listed in Column 3 were provided by the databases used, namely Agri-footprint, Eco-invent 3, ELCD, and Industry Data 2.0. The values in Column 4 were obtained from data collection in Column 3 and calculated based on the functional unit of 1 tonne of Nile tilapia fish harvested per year.

**Table 2.** Inputs parameters used in LCA at Nile tilapia cage farming.

| Activities (1) | Parameter Input (2) | Feed Ingredients (kg·t⁻¹) (3) | Quantity (4) |
|---|---|---|---|
| | | Fish meal (15%) | 309.34 kg |
| | | Meat meal (5%) | 103.11 kg |
| | | Soybean meal (22%) | 412.46 kg |
| **Feeding** | Formulation | Ground nut meal (10%) | 206.23 kg |
| | | Rice Bran (10%) | 206.23 kg |
| | | Wheat middling (16%) | 309.34 kg |
| | | Broken rice (16%) | 309.34 kg |
| | | fish oil (6%) | 82.49 kg |
| | | **Materials (kg·t⁻¹)** | |
| | | Netting (Nylon) | 406.81 kg |
| **Infrastructure** | Net Cage | Frame (Galvanized Steel) | 409.29 kg |
| | | High-density polyethylene (HDPE) drum | 450.22 kg |
| | | Boat (Glass fibre) | 8.96 kg |
| | | Boat Engine (Cast Iron) | 4.48 kg |
| **Operation** | Materials used | High-density polyethylene (HDPE) drum | 1.97 kg |
| | | Petrol | 33.07 kg |
| | | Electricity | 24.81 kWh |

**Table 3.** Outputs parameters used in LCA at Nile tilapia cage farming.

| Parameter Output (1) | Amount (kg·t⁻¹) (2) |
|---|---|
| Nitrous Oxide ($N_2O$) | 1.69 |
| Biochemical Oxygen Demand (BOD) | 5.328 |
| Chemical Oxygen Demand (COD) | 5.04 |
| Total Suspended Solid (TSS) | 100.8 |
| Ammonia-Nitrogen ($NH_3$-N) | 0.7344 |
| Phosphorus (P) | 0.72 |
| Nitrogen (N) | 0.36 |

The elements were determined based on the CML-IA method by the Centre of Environmental Science of Leiden University [11,21,29,30]. The value of nitrous oxide ($N_2O$) in Column 2 of Table 3 was calculated using Tier 1—Equation 4.10, sub section 4.3.2.1 of 2013 Supplement to the 2006 IPCC Guidelines for National Greenhouse Gas Inventories: Wetlands [31]. The values of for biochemical oxygen demand (BOD), chemical oxygen demand (COD), total suspended solid (TSS), ammonia-nitrogen ($NH_3$-N), phosphorus (P) and nitrogen (N) in Column 2 were obtained from laboratory results on water samples obtained from Kenyir Lake in 2018–2019 during the dry and wet seasons. All values in Column 2 were then calculated based on the functional unit of 1 tonne of tilapia fish harvested per year. The experimental data in this study was compared to a similar lake where no aquaculture takes place, as shown in Table 4.

**Table 4.** Experimental data from laboratory results.

| Parameter | Aquaculture Farm | No Aquaculture Activity |
|---|---|---|
| Biochemical Oxygen Demand (BOD) | 7.4 mg/L | 3.3 mg/L |
| Chemical Oxygen Demand (COD) | 7 mg/L | 6.5 mg/L |
| Total Suspended Solid (TSS) | 140 mg/L | 80 mg/L |
| Ammonia-Nitrogen (NH$_3$-N) | 1.02 mg/L | 4.5 mg/L |
| Phosphorus (P) | 0.9 mg/L | 0.2 mg/L |
| Nitrogen (N) | 1 mg/L | 0.4 mg/L |

*3.2. Effect on Various Environmental Impact Categories*

Figure 4 shows the environmental impact pattern of each category for Nile tilapia fish farming activities, namely feeding, infrastructure and operation.

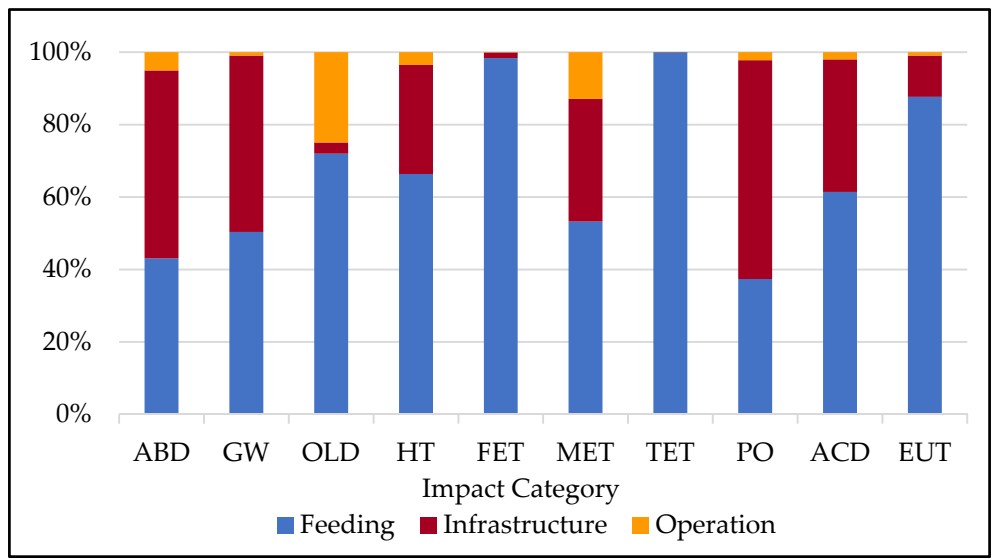

**Figure 4.** Effect on various environmental impact categories from Nile tilapia cage farming.

Based on the LCA study conducted, it was found that effects from ingredients in fish feed contributed an average of 55% to all the identified environmental impact categories. These findings are in line with other studies [19,27,32] which stated that feeding has the highest impact on the environment. Fish farming infrastructure contributed an average of 33%, while the effects from the operation of fish farming activities contributed an average of 12% to all the identified environmental impact categories, as shown in Figure 4.

3.2.1. Fish Feed

By referring to Figure 4, the effects from ingredients in fish feed were found to be at 43% for abiotic depletion (ABD), 50% for global warming (GW), 72% for ozone layer depletion (OLD), 66% for human toxicity (HT), 98% for freshwater ecotoxicity (FET), 53% for marine aquatic ecotoxicity (MET), 100% for terrestrial ecotoxicity (TET), 37% for photochemical oxidation (PO), 61% for acidification (ACD) and 88% for eutrophication (EUT).

The highest impact values from feed ingredients contributed to TET, due to fertilisers and pesticides used for agricultural ingredients in fish feed [33]. The effects from ingredients in fish feed were found to be the lowest for ABD, because the formulation of feed ingredients uses less fish meal,

as stated by Iribarren et al. [34]. It was also mentioned that the formulation of fish feed using more soybean than fish meal was expected to contribute less to ABD.

### 3.2.2. Infrastructure

The environmental effects of infrastructure were found to be at 52% for abiotic depletion (ABD), 49% for global warming (GW), 3% for ozone layer depletion (OLD), 30% for human toxicity (HT), 2% for freshwater ecotoxicity (FET), 34% for marine aquatic ecotoxicity (MET), 60% for photochemical oxidation (PO), 37% for acidification (ACD) and 11% for eutrophication (EUT) as shows in Figure 4.

The highest impact value from infrastructure at this study area was from MET because of the phosphorus and nitrogen released to the water [11]. These findings were in line with a previous study by Ayer and Tyedmers [35], where a high reading of MET was caused by netting. The lowest effect from infrastructure for fish farming activities was OLD, possibly due to the fact that the materials used in infrastructure did not contain chlorine or bromine.

### 3.2.3. Operation

As shown in Figure 4, the contribution of operation to the impact categories was generally lower than fish feed and infrastructure, namely 5% for abiotic depletion (ABD), 1% for global warming (GW), 25% for ozone layer depletion (OLD), 4% for human toxicity (HT), 13% for marine aquatic ecotoxicity (MET), 2% for photochemical oxidation (PO), 2% for acidification (ACD) and 1% for eutrophication (EUT).

Global warming was the highest value of all environmental categories for the impact of the operation of fish farming due to the presence of greenhouse gases and fuel consumption [18,19]. Fishing activities are commonly associated with the use of petroleum-based and diesel-burning boats, which have a high impact on GW.

### 3.3. Analysis on Feeding Ingredients

As mentioned by Silva et al. [36], fish feed production is the dominating process in all main categories of the aquaculture system. In this study, feeding was the main contributor in the aquaculture system as it represents the highest overall contribution in ten impact categories. The feed conversion rate (FCR) from our field observations was 2.1—FCR is the amount of feed required to produce a unit of fish. Since FCR in this study was more than 2, this is means 2 kg feed is required to produce 1 kg of fish. Hence, this contributed an impact to the environmental potential [24]. It was also mentioned that better feeding management should be implemented.

As observed in the results (Figure 5), impacts from broken rice were found to be at their highest for MET among all other feed ingredients, namely fish meal, rice bran, meat meal, wheat middling, soybean meal, broken rice, groundnut meal and fish oil. Paddy rice production has a high impact on MET due to the electricity needed for milling, fuel use for traction, production and transport [37]. Nutrient emissions from fish wastes and uneaten feed pellets located at the base of the lake also have an impact on MET due to phosphorus emissions [20].

Fish meal and soybean meal were the feed materials which contributed most to ABD and GW. Consistent with previous research [19], this fact is closely linked to the demand of great amounts of these materials according to the current fish feed formulation. Moreover, if global warming is of particular interest, additional hotspots would also include transport and the emissions to the air from boilers in feed processing [38]. On the other hand, emissions of N and P in the lake have the greatest impact on the environment, as mentioned by García et al. [19]. In contrast, fish feed formulation is said to be the most important contributor in fish production. Hence, the substitution of ingredients in feed formulation would be significant for reducing the environmental impacts resulting from fish feed [17,23].

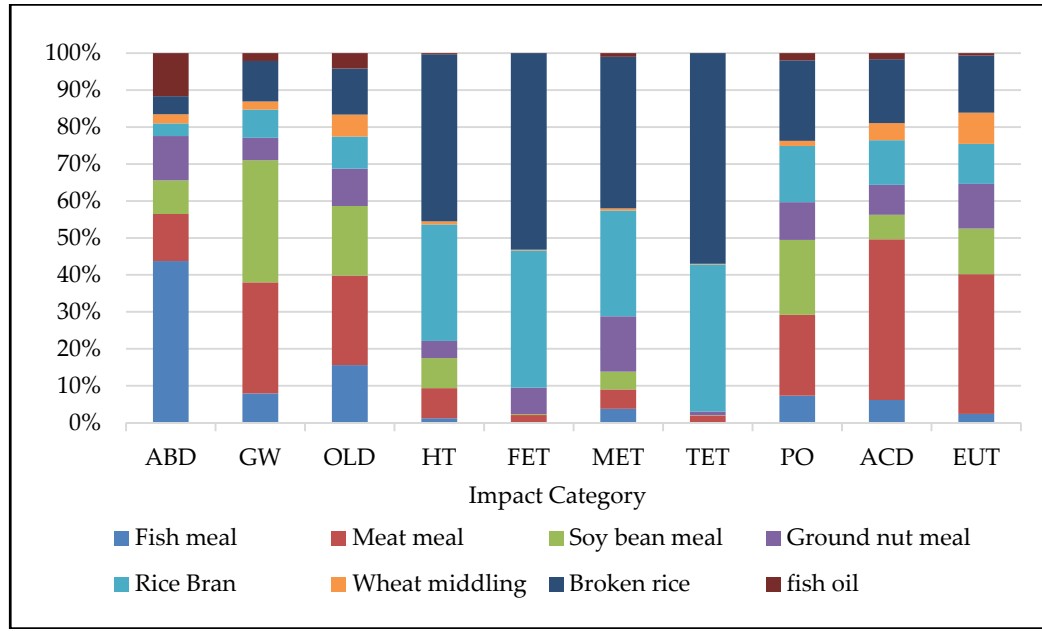

**Figure 5.** Contribution of feed composition to environmental impact potential.

### 3.4. Analysis on Infrastructure and Operation

The environmental characterisation of component materials from infrastructure and operation revealed which subsystems accounted for a greater potential impact. Figure 6 shows the percentage contribution of seven materials to potential environmental impacts. The contribution of the operation facilities to most of the impact categories was generally low, since the life expectancy of the materials is up to 20 years [39], except for GW and OLD, which basically comes from the utilisation of petrol. Petroleum burning involved in fishing activities is a common cause of GW and OLD. However, GW and OLD in this case study caused minimal impact compared to the feeding process.

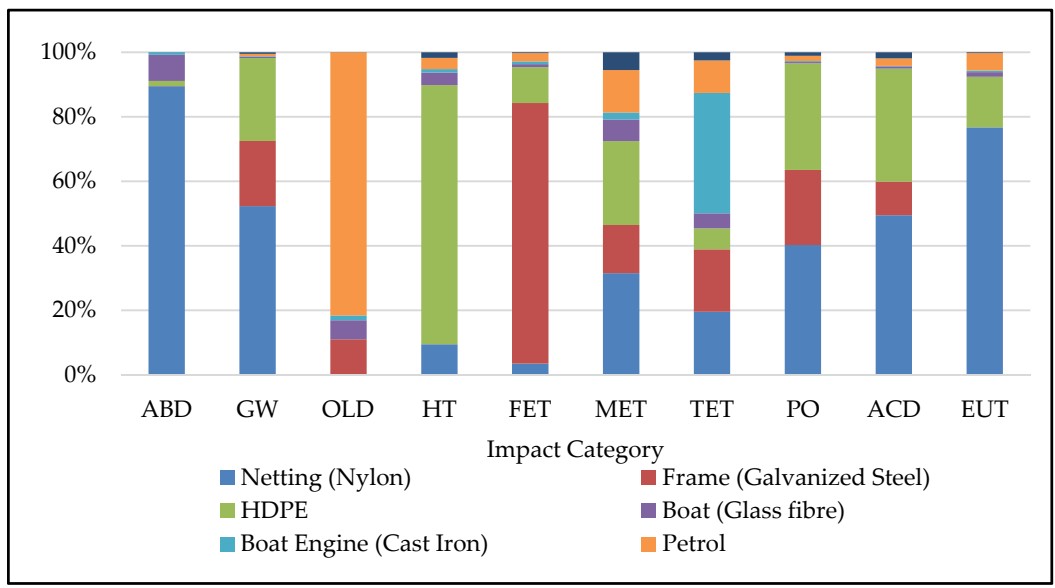

**Figure 6.** Contribution of infrastructures and operation materials to the environmental impact potential.

Infrastructure materials from netting (nylon), frames (galvanised steel) and high-density polyethylene (HDPE) represent the greatest contribution to ABD (90%), GW (52%), MET (32%), PO (40%), ACD (50%) and EUT (77%), while fish farm operation was responsible for the highest

contribution to GW from petrol and TET from boat engines. Having said that, netting made from nylon has the highest environmental impact, especially for ABD and EUT. Abiotic depletion and eutrophication can threaten the health of aquatic ecosystems, including aquaculture systems. Nevertheless, previous researchers assumed that infrastructure makes an inconsequential contribution to the impact on the life cycle of fish farming systems [32,34].

## 4. Sensitivity Analysis

A sensitivity analysis for the three most contributing parameters in the fish farming system was conducted to determine how important the data quality for each given parameter is, seeing as a highly sensitive parameter needs high quality data to achieve accurate results [35]. Sensitivity analysis is commonly provided by previous researchers to test for methodological choices and critical data sources [40,41]. Since there is no standard method for sensitivity analysis defined in ISO standards, an assumption of an alternative scenario was chosen to illustrate the different scenarios [40,42]. A factor of ± 1.25 was used in the analysis input of feeding, infrastructure and operation. The graph in Figure 7 shows the sensitivity for all categories in terms of environmental impact.

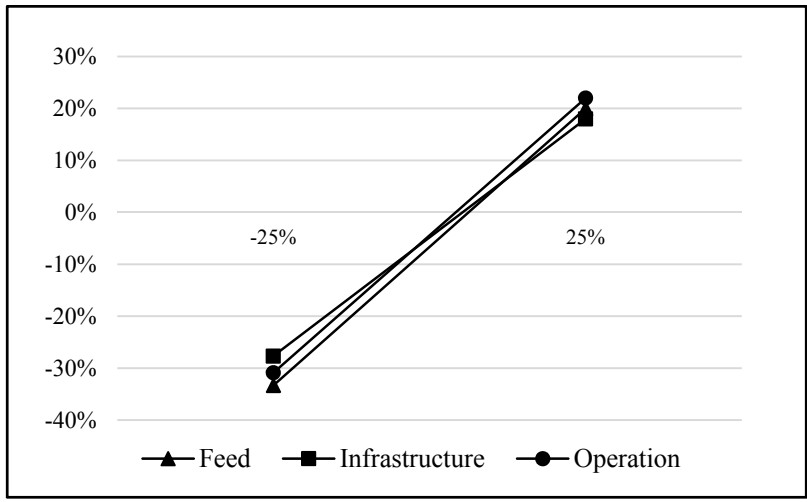

**Figure 7.** Uncertainty spider plot for the environmental impact potentials.

As mentioned earlier, the results show that the changes in input do affect the alternative scenarios compared to the base input. By reducing the input values by 25%, the percentages of the alternative scenarios was decreased by 33%, 28% and 31% for feed, infrastructure and operation, respectively. On the other hand, by increasing the input data by 25%, the alternative scenarios increased by 18%, 20% and 22% for feed, infrastructure and operation, respectively. This implies that the assumed parameters are in line with the present parameters. Therefore, it is possible to conclude that the study was reliable, but the results must be adapted with caution.

## 5. Conclusions and Recommendation

LCA proved to be a useful tool for assessing the environmental performance of Nile tilapia aquaculture farming. Chain transparency and accountability were among the benefits of the use of LCA in this case study. Moreover, the environmental hot spots within fish farming were identified. As stated in other aquaculture systems of fish farming, fish feeding operations and emissions of N and P due to the metabolism of the species are factors that have a significant impact on the environment. Having said that, raw materials from infrastructure and operation could be negligible, as they have less environmental impact. Recommendations for feed formulation manufacturers are centred on raw material production. Thus, different raw materials and/or ingredient ratios should be assessed as a mitigation plan to sustain the environmental health of the lake. The LCA performed for Nile

tilapia farming is useful for its implementation in the study of the environmental performance of aquaculture plants.

**Author Contributions:** The first author is involved in conducting the study onsite with assistance from the second and third authors by giving advices and recommendation to further improved the research. The three authors are actively involved in drafting the manuscript. All authors have read and agreed to the published version of the manuscript.

**Funding:** This research and the APC was funded by TNB Seeding Fund, grant number [U-TG-RD-19-06].

**Acknowledgments:** The authors would like to acknowledge TNB Seeding Fund (U-TG-RD-19-06), UNITEN R & D Sdn. Bhd. Universiti Tenaga Nasional, Malaysia and Faculty of Science and Technology, Universiti Kebangsaan Malaysia for technical and financial support.

**Conflicts of Interest:** The authors declare no conflict of interest.

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
