# Peer review of "Life Cycle Assessment of Nile Tilapia (Oreochromis niloticus) Farming in Kenyir Lake, Terengganu"

_sustainability, doi:10.3390/su12062268_

Round 1

Reviewer 1 Report

General comments: the analytical method needs to be elaborated. Please provide a more detail explanation of your research methods. The procedure for collecting data should be specific and descriptive information on sample farms need to be reported.

1) Since feeds differ in composition and content depending on the species, specific information on the feed used should be provided. In particular, tilapia is a herbivore fish and needs relatively low fish meal contents.

2) Feed conversion rate (FCR) of sample fish farms and the mortality rate of sample aquaculture products, as well as stocking density for tilapia farming in the region, should be considered in your analysis.

3) Finally, Figure 1's resolution should be improved.

Reviewer 2 Report

The manuscript is well written and the results are clear. I included some revisions in the attached PDF file.

Round 2

Reviewer 1 Report

General comments: your paper has improved. However, you haven’t resolved some issues.

Author’s response 1: The analytical methods, explanation of the method and procedure for data collection have been provided in the revised manuscript in the methodology section.

Reviewer’s response 1: As I mentioned before, please provide detailed information on sample farms like location (with a map), sampling frequency, sampling date and period, etc. Additionally, fish production operations should be specified because you can separate the phase of production, calculating the environmental impacts by the stage. Fish farming is not a simple job.

Response 2: The percentage of fish feed composition in section 3.1 were obtained from the fish feed supplier, while the quantity was calculated based on the usage per tonne harvested fish.

Reviewer’s response 2: If you got the information about fish feed composition from the fish feed supplier, please explain it specifically in the manuscript. As I said before, the fish feed and ingredients and farming period could be different at the farming stages (Fry, Fingerling, and Growout). You should consider it.

Reviewer’s point 3: Feed conversion rate (FCR) of sample fish farms and the mortality rate of sample aquaculture products, as well as stocking density for tilapia farming in the region, should be considered in your analysis.

Authors’ response3: The factors mentioned above have been incorporated in the analysis.

Reviewer’s response 3: The reviewer could not find such information (FCR and mortality rate) in your paper. The amount of feed consumption depends on such factors. If you considered it, please provide the information and how to consider it in your analysis. To get a more realistic conclusion, you should consider such information. It is crucial. Please resolve this issue.

Reviewer 2 Report

The manuscript is well revised and can be accepted in its current form.

Author Response

Thank you so much for the comments.

Round 3

Reviewer 1 Report

 All issues have been resolved.

This manuscript is a resubmission of an earlier submission. The following is a list of the peer review reports and author responses from that submission.

Round 1

Reviewer 1 Report

Letter to Authors,
sustainability-610151
Life Cycle Assessment of Nile Tilapia (Oreochromis niloticus) Farming in Kenyir Lake, Terengganu
Hayana Dullah, M. A. Malek, Marlia M. Hanafiah

190923

Dear authors,
I am upset reading your thoughtless MS again with thoughtless revision. You seem at a front of destroying tropical fresh and brackish-water biodiversity. Unless you give statements regarding needs for developing physical or biological containment of the highly invasive fish in the culture cages or eradication measures in the wild, publishing your report is adverse to the journal's scope. Your neglect of the primary issue of tilapia's invasiveness indicates it was hopeless to resubmit.
I am sorry for this disappointing review result.

Reference

Arthur RI, Lorenzen K, Homekingkeo P, Sidavong K, Sengvilaikham B, Garaway CJ, 2010. Assessing impacts of introduced aquaculture species on native fish communities: Nile tilapia and major carps in SE Asian freshwaters. Aquaculture 299:81-88.

Canonico GC, Arthington A, McCrary JK, Thieme ML. 2005. The effects of introduced tilapias on native biodiversity. Aquat Conserv Mar Freshw Ecosyst 15:463-483.

Deines AM, Wittmann ME, Deines JM, Lodge DM. 2016. Tradeoffs among ecosystem services associated with global tilapia introductions. Rev Fish Sci Aquacult 24:178-191.

Ortega JCG, Julio HF, Gomes LC, Agostinho AA. 2015. Fish farming as the main driver of fish introductions in Neotropical reservoirs. Hydrobiologia 746:147-158.

Reviewer 2 Report

The manuscript has been improved, but still demands some work, especially regarding correct English. Some specific comments are proposed directly on the review pdf (attachment).
